# Population genomic and historical analysis suggests a global invasion by bridgehead processes in *Mimulus guttatus*

Mario Vallejo-Marín [1✉], Jannice Friedman [2], Alex D. Twyford[3,4], Olivier Lepais[5], Stefanie M. Ickert-Bond[6], Matthew A. Streisfeld[7], Levi Yant [8], Mark van Kleunen [9,10], Michael C. Rotter[11] & Joshua R. Puzey[12]

Imperfect historical records and complex demographic histories present challenges for reconstructing the history of biological invasions. Here, we combine historical records, extensive worldwide and genome-wide sampling, and demographic analyses to investigate the global invasion of *Mimulus guttatus* from North America to Europe and the Southwest Pacific. By sampling 521 plants from 158 native and introduced populations genotyped at >44,000 loci, we determined that invasive *M. guttatus* was first likely introduced to the British Isles from the Aleutian Islands (Alaska), followed by admixture from multiple parts of the native range. We hypothesise that populations in the British Isles then served as a bridgehead for vanguard invasions worldwide. Our results emphasise the highly admixed nature of introduced *M. guttatus* and demonstrate the potential of introduced populations to serve as sources of secondary admixture, producing novel hybrids. Unravelling the history of biological invasions provides a starting point to understand how invasive populations adapt to novel environments.

[1] Biological and Environmental Sciences, University of Stirling, Stirling, Scotland, UK. [2] Biology Department, Queen's University, Kingston, ON, Canada. [3] School of Biological Sciences, University of Edinburgh, Edinburgh, Scotland, UK. [4] Royal Botanic Garden Edinburgh, Edinburgh, UK. [5] INRAE, Univ. Bordeaux, BIOGECO, Cestas, France. [6] Herbarium (ALA), University of Alaska Museum of the North, University of Alaska Fairbanks, Fairbanks, AK, USA. [7] Institute of Ecology and Evolution, University of Oregon, Eugene, OR, USA. [8] Future Food Beacon and School of Life Sciences, University of Nottingham, Nottingham, UK. [9] Department of Biology, University of Konstanz, Konstanz, Germany. [10] Zhejiang Provincial Key Laboratory of Plant Evolutionary Ecology and Conservation, Taizhou University, Taizhou, China. [11] Department of Biological Sciences, Northern Arizona University, Flagstaff, AZ, USA. [12] Biology Department, College of William and Mary, Williamsburg, VA, USA. ✉email: mario.vallejo@stir.ac.uk

Increasing global connectivity is leading to widespread species translocations[1]. Most biological communities now include introduced members that have recently moved beyond their native ranges, often with negative impacts[2–6]. Finding the origins of invaders helps develop strategies for prevention, management and eradication[7,8]. It is also crucial for understanding to what extent invaders adapted to novel environments, along with the mechanisms of such adaptations[9,10].

Tracing the migration and spread of invasives is typically very challenging. Inferring introduction histories is often accomplished using historical records, genetic analyses, or a combination of both[11–13]. In most cases, historical records of first introduction are unavailable or unreliable. Genetic data has greatly improved our ability to study the origins of invasions, and often uses information derived from extant populations[10]. However, genetic inferences are usually confounded by demographic processes that shape the introduced populations, including multiple introduction events, bottlenecks, evolution in the introduced range, admixture and hybridisation[13–15].

Here, we use historical and genomic data to generate and test hypotheses in order to unravel the rapid worldwide invasion by the common yellow monkeyflower, *Mimulus guttatus* Fischer ex DC. (*Erythranthe spp.* (L.) G. L. Nesom; Phrymaceae), a herbaceous plant native to Western North America that was introduced across the world in the nineteenth century[16–20]. Unlike many invasive and non-native species, detailed historic botanical records[21] and travel diaries of early explorers[22] allow us to clearly retrace the history of the first introduction of *M. guttatus* into Europe. Historical records of *M. guttatus* reaching the United Kingdom (UK) paint a clear picture, but beyond this little is known. Here, we consider the hypothesis that the UK acted as a bridgehead (where invasive populations serve themselves as sources for further invasions[12,23]) for worldwide invasion.

The first European record of *M. guttatus* appears in *Curtis's Botanical Magazine*[21], which presents a plate of *Langsdorff's Mimulus* (*Mimulus langsdorfii* Donn ex Sims), featuring a flowering individual of *M. guttatus*. The provenance of the depicted material is from Grigori von Langsdorff who "…brought it, as we are informed, from Unalashka, one of the Fox Islands" (Unalaska, Aleutian Islands)[21], in his capacity as a naturalist on a Russian expedition to the Alaskan territories in 1805. Langsdorff describes how the expedition reaches Unalaska on 16 July 1805, and, after anchoring in Sea-Otters Bay (probably present-day Ugadaga Bay), they travelled on foot to Iluluk (Dutch Harbor). Here, Langsdorff first encounters *M. guttatus*: "splendid flowers were in blow upon the shore, among which a new Mimulus and Potentilla, which has never yet been described, were particularly to be distinguished".[22] (p. 329). Material brought by Langsdorff made its way to various Botanic Gardens, including Moscow (where it is listed as *M. guttatus* Fischer *nom. nudum*) and Montpellier (where De Candolle validly published the name *M. guttatus*). The seeds of *M. guttatus* also reached the Botanic Gardens at Cambridge in 1812, and it is therefore almost certain that the original species description included specimens collected by Langsdorff in Unalaska[19].

Presciently, the *Botanical Magazine* recognised the potential for *M. guttatus* to become established outside western North America, and the 1812 entry states that because the taxon has showy flowers and is "easily propagated by seeds, and most probably by its runners, must soon be very common."[21]. In fact, the first naturalised populations in the British Isles are recorded by 1830[24], rapidly spreading throughout the UK[25]. The introduction history of *M. guttatus* outside of the UK is much less well understood. *Mimulus guttatus* seems to have reached New Zealand and become naturalised by 1878[26], and the introduction of this taxon to eastern North America may have occurred much

later in the second half of the twentieth century[27]. Therefore, the material brought in by Langsdorff represents the first introduction of *M. guttatus* outside its native range, and the subsequent arrival and naturalisation on the British Isles is the best documented, and currently most widespread, monkeyflower invasion[17,20,24,25,28,29].

The historical hypothesis of an Alaskan origin of European monkeyflowers is consistent with results from previous genetic analysis of *M. guttatus* in the United Kingdom[30,31]. However, these studies did not include material from the putative origin (Aleutian Islands), and due to their focus on UK populations, did not examine genetic relationships between native populations and introduced populations in other parts of the range such as in Eastern North America, the Faroe Islands, mainland Europe and New Zealand. Native *M. guttatus* presents an enormous breadth of ecological and genetic diversity[32,33], and it remains unknown how much of this diversity is represented among introduced populations and the extent to which non-native populations have diverged. Recently, Da Re et al.[20] used climatic niche modelling to compare the climatic envelope of native and introduced *M. guttatus* populations, finding no evidence of niche shift in the introduced UK populations compared to the native ones. Moreover, the highest niche similarity of invasive UK populations occurred in the Aleutian Islands[20], lending support to the historical hypothesis that traces their origin to Langsdorff.

Here, we provide the first global genetic analysis of native and introduced populations of *M. guttatus* by marrying historical information with genomic analyses. Specifically, we: (1) Resolve range-wide relationships at the population-level in the introduced range, as well as in the native range including the previously under-sampled regions of the Aleutian Islands and mainland Alaska; and (2) use genomic data to reconstruct the population genetic history of introduced UK populations and test the hypothesis that UK populations have a simple Aleutian origin or are the product of a more complex invasion history. Our results show that populations from the British Isles were first likely introduced from the Aleutian Islands in Alaska, followed by rapid admixture from other parts of the native range. We hypothesise that these British populations then served as a bridgehead for invasions worldwide into the rest of Europe, New Zealand and eastern North America. Our findings emphasise the highly admixed nature of introduced *M. guttatus* and raise the possibility that introduced populations might serve as sources of secondary admixture, producing novel hybrids.

## Results

**Population relationships in the native range.** The global sampling of *M. guttatus*, including populations sampled across ~5000 km of its distribution in North America (Fig. 1), allowed us to resolve population groupings in both native and introduced ranges. In the native range, including the newly sampled Alaskan region, strong geographic structure is evident from phylogenetic analysis (Fig. 2), with four well-resolved North, South, Coastal and North Pacific clades[34]. The newly sampled populations in Alaska and the Aleutian Islands form part of the North Pacific Clade (Fig. 2). This clade is sister to the Coastal clade and includes populations from northern Washington to the westernmost Aleutian Islands (Attu Island). Phylogenetic analysis revealed an unexpected placement of some populations from inland Oregon, including those from Iron Mountain, which conflicts with previous analyses and their expected relationships based on simple geography. The tetraploid *M. guttatus* population sampled in the Shetland Islands in the UK is nested among other geographically proximate populations, further supporting the local origin of this autopolyploid in the introduced range[35].

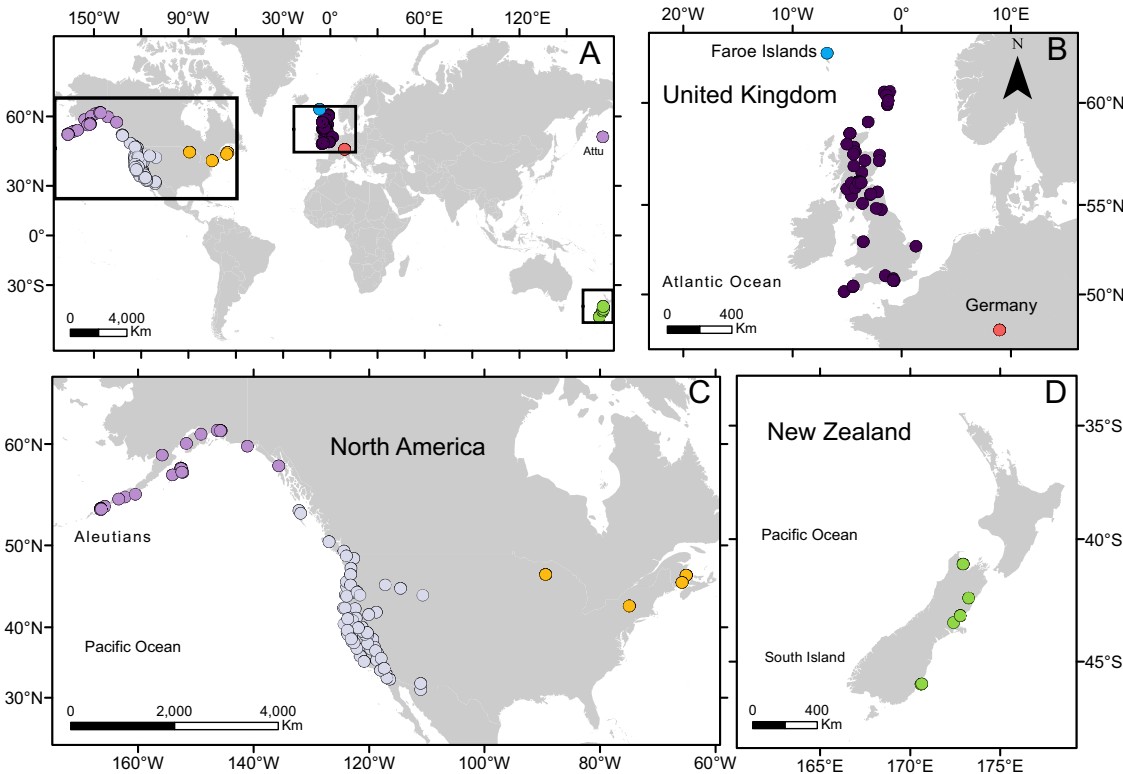

**Fig. 1 Global sampling of *Mimulus guttatus* populations. A** Overview of all sampled population. **B** European populations. **C** North American populations. Populations in the west are native, while orange symbols represent invasive populations in eastern North America. **D** New Zealand populations. The symbols for the two southernmost populations in the South Island are overlapping.

Finally, *M. luteus* formed a strongly supported clade, and the triploid and allohexaploid hybrids, *M x robertsii and M. peregrinus* can be clearly distinguished from both parental taxa (*M. guttatus* and *M. luteus*).

**Global invasion of *Mimulus guttatus*.** At a global scale (Fig. 1), introduced *M. guttatus* populations are scattered across the phylogeny, indicating many independent introductions from across the native range (Fig. 2). In contrast, all UK *M. guttatus* populations form a sister group to the North Pacific clade. The UK group also includes other non-native populations from New Zealand, Canada and Germany, suggesting it may be the source for these. Other New Zealand populations are grouped within the Coastal clade, suggesting a potential second introduction. Moreover, interesting geographic discontinuities exist in North America, with a non-native New York population nested in the native North clade. Finally, two additional populations from eastern North America, as well as the single sampled population from the Faroe Islands are grouped together with the native HAM-SWC group from Oregon (Fig. 2). Thus, the UK populations are genetically similar to each other and are closely related to some of the introduced populations of *M. guttatus* in New Zealand and eastern North America. However, the placement of other non-native populations within various native clades clearly indicates additional, independent introductions to New Zealand, eastern North America and the Faroe Islands, suggesting a complex history of colonisation.

Among native populations, those from the UK form a separate genetic cluster, as seen in principal component analysis (PCA) (Fig. 3). As in the phylogenetic reconstruction, the UK group is closely associated with non-native populations from New Zealand, Germany and eastern North America. The PCA is

also consistent with two separate introductions into New Zealand, one of them closely related to UK populations, and three independent origins of non-native populations in eastern North America. One of these origins of eastern North American populations is shared with the population from the Faroe Islands, forming a distinct group with two native populations from Oregon (SWC and HAM; Fig. 3). An interactive version of Fig. 3 with labelled individuals and populations is available at https://mvallejo6.github.io/mimulus_voyage. Population structure in the native range is less clear from the worldwide PCA, although the North Pacific clade and particularly the Aleutian Islands populations are well differentiated along the first principal component (Fig. 3).

Worldwide groupings by *K*-means cluster analysis (Fig. 4 and Supplementary Fig. 1) partition North American samples into three groups, New Zealand into two groups, and the single populations from the Faroe Islands and Germany in one group each, largely consistent with the results above. Non-native UK populations form two groups, one mixed with European and Eastern North American samples, and another with New Zealand samples. Native, non-Alaskan populations are distributed in five groups. Aleutian populations form a separate group not shared with other geographic regions. The *fastStructure* analysis with $K^* = 7$ and $K = 8$ (Supplementary Fig. 2) provides further support for these groupings ($K^* =$ optimal K estimated in ref. [36]). UK populations form a separate group with multiple affinities to New Zealand and eastern North American samples. Furthermore, the distinctiveness of Aleutian populations relative to other native populations is also obvious (e.g., cluster 4 at $K = 8$, Supplementary Fig. 2). In terms of nucleotide diversity ($\pi_{GENOME}$), UK populations show, on average, a relatively high level of diversity compared to Alaskan populations ($\pi_{GENOME} = 0.0047$, $n = 43$ populations vs. $\pi_{GENOME} = 0.0037$, $n = 32$, respectively), but only slightly lower than the overall average

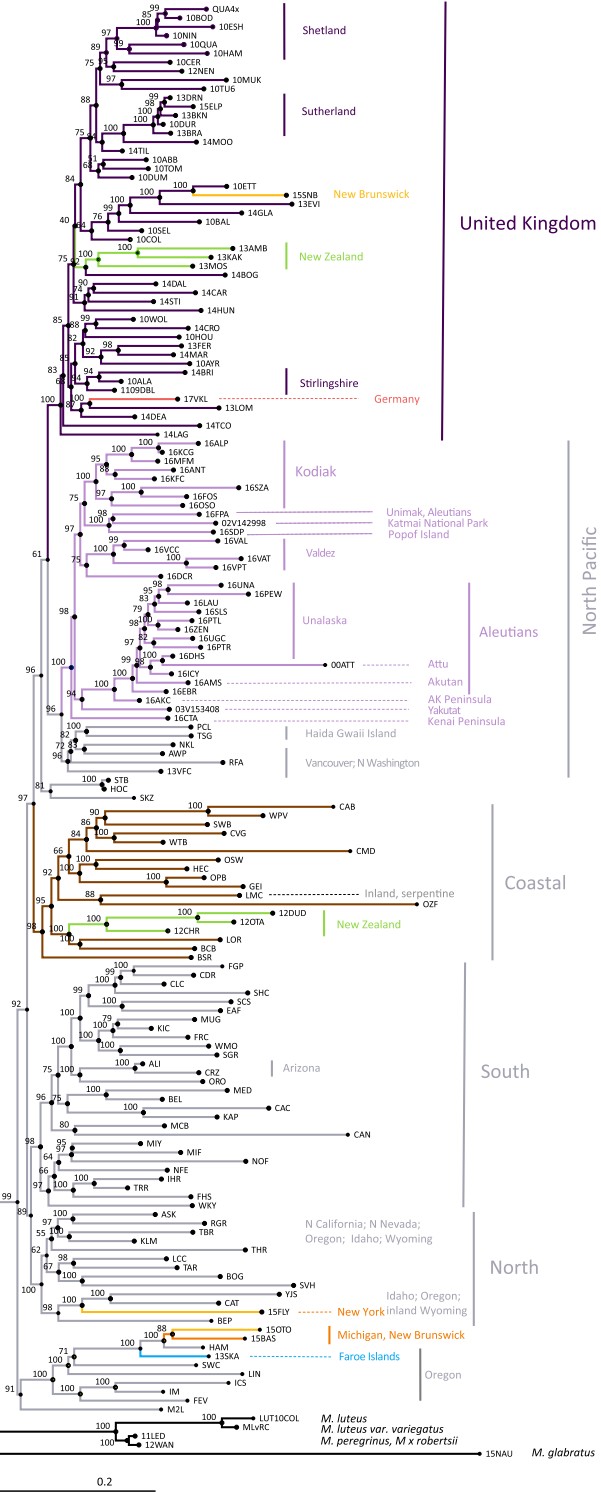

**Fig. 2 Phylogenetic reconstruction of studied *Mimulus*.** Maximum likelihood phylogenetic reconstruction of the relationship between studied *Mimulus guttatus* populations, and including populations from *M. luteus* (LUT10COL, UK), *M. luteus* var. *variegatus* (MLvRC, Chile) *M x robertsii* (12WAN) and *M. peregrinus* (11LED). The tree is rooted using a population of *M. glabratus* from Michigan (15NAU).

of other native North American populations ($\pi_{GENOME} = 0.0050$, $n = 68$ populations; Supplementary Fig. 5). Other invasive populations have similar or lower average levels of nucleotide diversity as the UK ($\pi_{GENOME} = 0.0031$–$0.0047$), although the number of

sampled populations in each of these regions is small (1–6; Supplementary Fig. 5).

**Introduction history in the UK**. To estimate a most likely scenario for the origin and history of introduction of UK populations, we next performed a coalescent analysis with ABC. Our analysis of demographic models allowed us to compare different scenarios for the origin and history of introduction of UK populations relative to five genetic groups in the native range: Aleutians (ALE) and Alaska-British Columbia (AKBC), both of which form part of the North Pacific clade, and the North (NORTH), South (SOUTH), and Coastal (COAST) clades (see Figs. 2 and 4)[34,37]. When assuming a single introduction event, the most likely source of UK individuals is the AKBC group (Table 1, posterior probability $p = 0.89$). However, model comparisons favour scenarios with additional waves of introductions (Table 1). When we model two introductions, a first introduction from AKBC followed by a second introduction wave from NORTH has greatest support (Table 1, $p = 0.48$) and is more likely than a single introduction scenario (237 votes against 32 votes, Table 1). Similarly, three introduction models result in selecting an introduction history with a first introduction from AKBC followed by additional introductions from NORTH and COAST ($p = 0.53$, Table 1) and then four introduction models identify a first introduction from SOUTH followed by additional introductions from AKBC, NORTH and COAST as the most likely scenario ($p = 0.55$, Table 1). Finally, when comparing all best one- to four-wave introduction models, with all possible five-wave introduction models, the most likely introduction history identified consisted of a first introduction from ALE followed by four subsequent waves from the AKBC, NORTH, SOUTH and COAST (E4 model; $p = 0.55$, Table 1). Full demographic parameters (e.g., estimated population sizes and introduction times per genetic group; E4 model) are presented in Supplementary Table 1.

Classification of the datasets simulated under the best one- to five-wave introduction scenario showed that 41.6% of the simulated datasets under E4 scenario were correctly classified, and a 50.6% probability that a simulation classified as E4 truly originated from this model (Table 2). Thus, the combination of the type and number of molecular markers and model prior specifications we used here contain enough information to confidently differentiate scenarios with different number of introductions (e.g., single or two-wave introductions vs. three- to five-wave introductions). Nevertheless, distinguishing the most likely scenario among these complex and sometimes very similar multiple introductions scenarios proved more difficult (Table 2, Supporting Material File 1). This is likely due to the wide prior distribution specification for the proportion of migrants that ranged from 0.001 to 0.999 (Supporting Material File 1). A scenario with a five-waves introduction but with very few migrants originating from one of the source (for instance 0.2% of the receiving population) would be very similar to a four-wave introduction. Conversely, a five-waves scenario with a very high migration rate from one source and very low migration rate from the other sources would underestimate the number of introductions. Given the scenario complexity, our ability to distinguish the order of introductions is even more limited. The combination of relatively low posterior probability of the best scenario (0.55, Table 1) and high prior error rate (0.41, estimated from out-of-bag procedure) indicates moderate levels of confidence in scenario choice that is typical for complex multiple introductions scenario[38].

The posterior probability of 0.55 for the E4 model (Table 1), supports a first introduction from ALE followed by additional

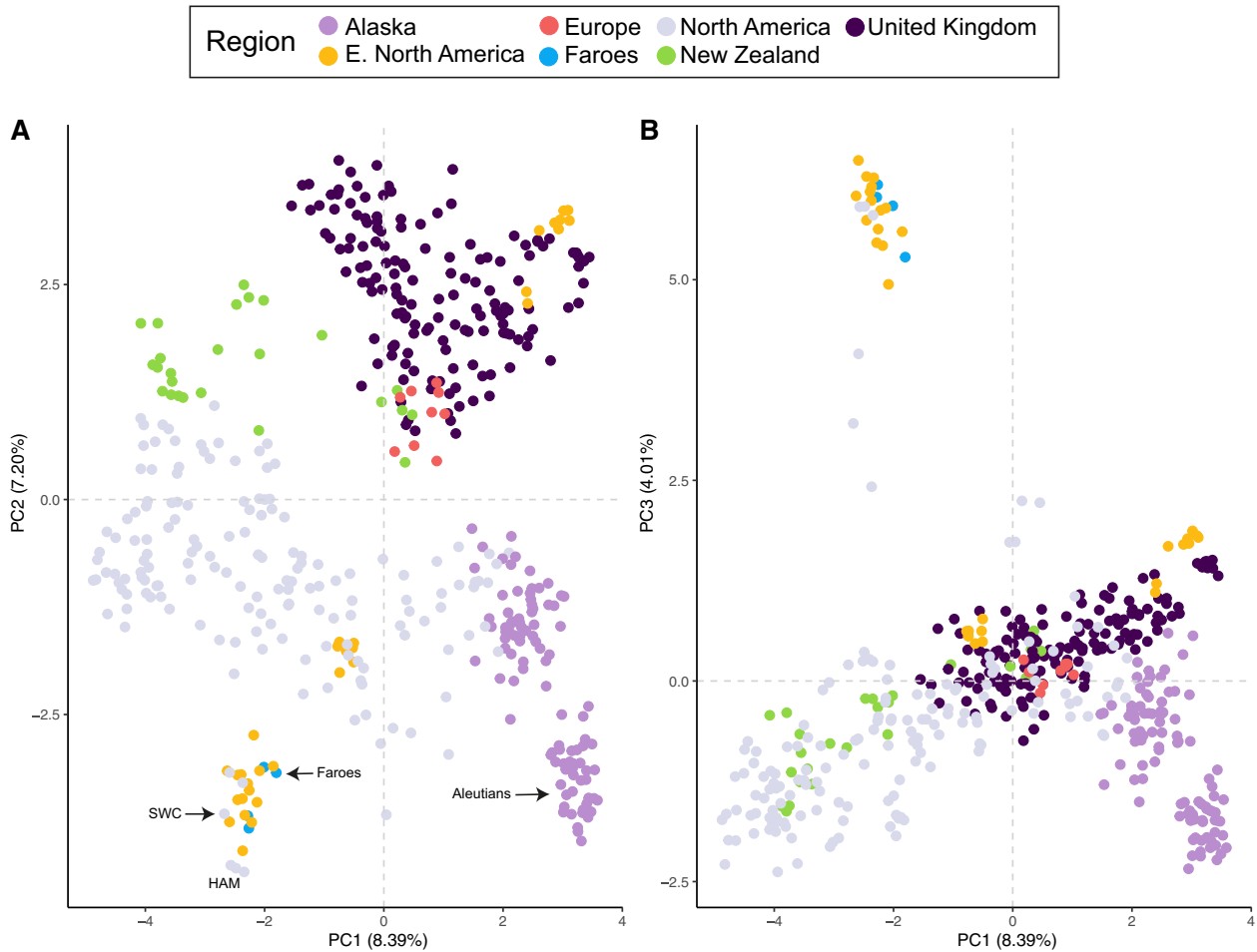

**Fig. 3 Principal component analyses of native and introduced _Mimulus guttatus_.** Principal component analysis (PCA) of 474 individuals of _Mimulus guttatus_ from both native and introduced populations genotyped at 1498 binary SNP loci. **A** Scatterplot of the first two principal components (PC2 vs. PC1). **B** Scatterplot of first and third principal components (PC3 vs. PC1). Colours indicate sample regions. An interactive 3D figure with individually labelled data points is available at: https://mvallejo6.github.io/mimulus_voyage/. A visualisation of the first 200 principal components (cumulative variance explained = 90.9%) projected in two dimensions using Uniform Manifold Approximation and Projection for Dimension Reduction (UMAP)[77] is shown in Supplementary Fig. 4.

introductions from the other four origins (Supplementary Fig. 3). However, most of the posterior distributions of demographic parameters (e.g., effective population size, number of generations since introduction) for model E4 were nearly identical to the prior distributions (Supplementary Table 1 and Supplementary Notes 2), indicating limited information content of the genetic data set to estimate the demographic parameters of this complex introduction history.

## Discussion

Here, we provide the first global picture of the genetic relationships between native and introduced populations of _Mimulus guttatus_, including targeted sampling of a historically indicated origin for the UK populations. Our results can be summarised in three main findings: (1) _Mimulus guttatus_ achieved a broad distribution across geographic boundaries through multiple introductions from genetically distinct source populations. (2) In some cases, the establishment of _M. guttatus_ in the invasive range might have been achieved via a bridgehead process, where invasive populations serve themselves as sources for further, more distant vanguard invasions. This is well illustrated in our discovery of the establishment of invasive populations in New Zealand and eastern North America by way of UK invasive populations. (3) Admixture in the introduced range has given rise to genetically distinct

populations generating novel genetic, and therefore phenotypic, combinations (e.g., invasive phenotypes that produce both large numbers of flowers and are highly clonal[39]).

Widely distributed taxa that serve as a source of invasive populations pose a particular challenge for molecular studies aiming to reconstruct the history of biological invasions. The distribution of _M. guttatus_ spans from Mexico to the Aleutians and covers >6000 km of coastline[33]. To identify potential sources of specific invasion events, sampling large geographic regions is required. _Mimulus guttatus_ has been the subject of continuous study for the last 60 years[32], and previous work has collected population samples across nearly its entire native range[40–42]. Our analyses of large-scale population samples from the native range builds on the recent finding of geographic genetic structure corresponding to separate coastal and northern colonisation events in North America[37]. Here, we fill in crucial gaps with sampling from Alaska and the Aleutian Islands, which reveals strong geographic structure in the far north west of the species range, with genetic clusters by islands in the Aleutians. This extensive sampling in the native range allows us to show that Aleutian populations have acted as important conduits to the invasion of _Mimulus_ in Europe and beyond.

Many biological invasions by both plants and animals are associated with multiple introductions, to the extent that single

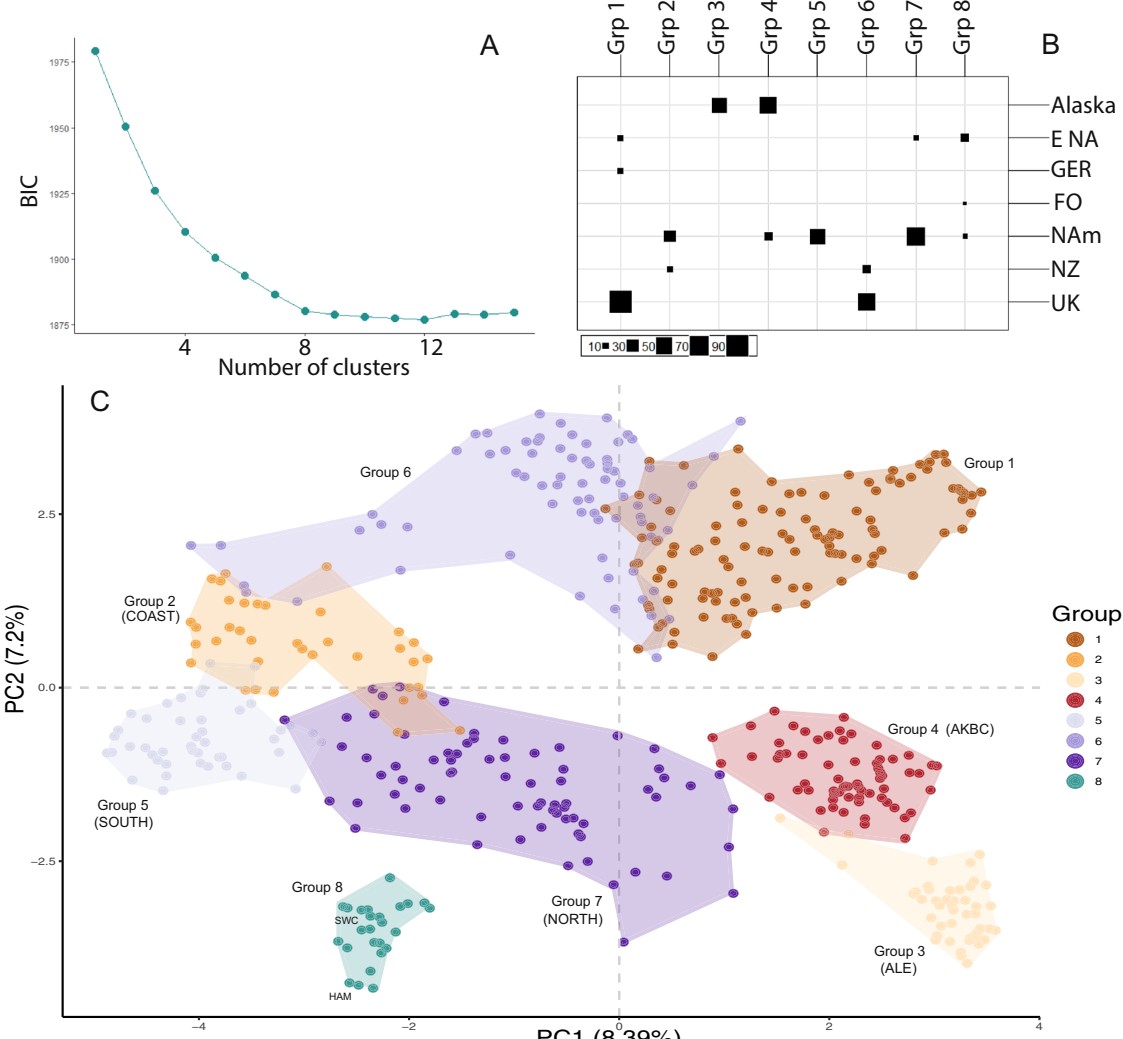

**Fig. 4 Clustering analysis of native and introduced populations of *Mimulus guttatus*.** *K*-means clustering analysis of native and introduced populations of *Mimulus guttatus*. The analysis is based on the first 300 principal components. **A** Bayesian Information Criterion values for models ranging from 2 to 15 clusters. **B** Group membership of each geographic group for the optimal number of clusters (*K* = 8). **C** Principal component analysis depicted in Fig. 3A but coloured by the groups identified in the *K*-means cluster analysis (*K* = 8). To improve clarity a polygon was drawn around each of the eight genetic groups. The genetic groups of native populations used for the ABC analysis are named in parenthesis: AKBC = North Pacific Clade.

introduction invasions are considered the exception[9]. Here, we found clear evidence that introduction of *M. guttatus* into various geographic regions has occurred by colonisation from multiple genetically distinct sources. For example, among the four populations we sampled in eastern North America, where *M. guttatus* was introduced in the last century, there is evidence of three genetically distinct groups, one of which also occurs in the Faroe Islands (Fig. 3). Similarly, introduced populations in New Zealand have at least two separate genetic origins, including a close affinity with native populations (near Santa Cruz, California) located 11,000 km away and with non-native populations in the UK. The multiple origins of invasive populations found in the same geographic region is important for several reasons. From a management perspective, multiple introductions can help identify locations of transport routes that are susceptible for further invasions. Moreover, multiple introductions may help invasive populations overcome demographic and genetic bottlenecks associated with introduction events[9], as demonstrated by the relatively high levels of nucleotide diversity observed in introduced populations of *M. guttatus* in the British Isles (Supplementary Fig. 5). In species that are introduced via the ornamental

trade, as was probably the case for monkeyflowers, repeated introductions may not be unusual. To date it is still possible to freely purchase monkeyflowers in UK garden centres. However, because the type sold is no longer *M. guttatus* but horticultural varieties of its close relative *M. luteus*, we speculate that the multiple introductions detected in the invasive range of *M. guttatus* reflect historical events (nineteenth and twentieth centuries) rather than recent reintroductions. In addition, we did not find evidence of large-scale admixture from *M. luteus* shaping genetic variation in *M. guttatus*, consistent with the strong reproductive barriers imposed by differences in ploidy level between these *Mimulus* taxa[43].

The genetic history of these invasions reveals a complex series of introduction events associated with early establishment (nineteenth century). Our ABC analyses reconstruct this history and show that extant populations are composed of a combination of multiple genetic groups from across the native range. Reconstruction of demographic events during introduction (Supplementary Fig. 3) supports an initial introduction of *M. guttatus* from the Aleutian Islands, which is consistent with the historical records of Langsdorff's expedition and subsequent transfer of

**Table 1 Comparison of demographic models of the invasion of *Mimulus guttatus* into the United Kingdom.**

| Model group | Number of introduced origins | Model | First introduced origin | Following introduced origins | Votes (posterior probability of best model) |
|---|---|---|---|---|---|
| A | 1 | A1 | ALE | | 121 |
| | 1 | **A2** | **AKBC** | | **276 (0.89)** |
| | 1 | A3 | COAST | | 219 |
| | 1 | A4 | NORTH | | 206 |
| | 1 | A5 | SOUTH | | 178 |
| B | 1 | A2 | AKBC | | 32 |
| | 2 | B1 | AKBC | ALE | 45 |
| | 2 | B2 | AKBC | COAST | 78 |
| | 2 | **B3** | **AKBC** | **NORTH** | **237 (0.48)** |
| | 2 | B4 | AKBC | SOUTH | 172 |
| | 2 | B5 | ALE | AKBC | 30 |
| | 2 | B6 | COAST | AKBC | 92 |
| | 2 | B7 | NORTH | AKBC | 183 |
| | 2 | B8 | SOUTH | AKBC | 131 |
| C | 1 | A2 | AKBC | | 28 |
| | 2 | B3 | AKBC | NORTH | 30 |
| | 3 | C1 | AKBC | NORTH,ALE | 60 |
| | 3 | **C2** | **AKBC** | **NORTH,COAST** | **160 (0.53)** |
| | 3 | C3 | AKBC | NORTH,SOUTH | 74 |
| | 3 | C4 | NORTH | AKBC,ALE | 98 |
| | 3 | C5 | NORTH | AKBC,COAST | 118 |
| | 3 | C6 | NORTH | AKBC,SOUTH | 114 |
| | 3 | C7 | ALE | AKBC,NORTH | 96 |
| | 3 | C8 | COAST | AKBC,NORTH | 136 |
| | 3 | C9 | SOUTH | AKBC,NORTH | 86 |
| D | 1 | A2 | AKBC | | 22 |
| | 2 | B3 | AKBC | NORTH | 34 |
| | 3 | C2 | AKBC | NORTH,COAST | 106 |
| | 4 | D1 | AKBC | NORTH,COAST,ALE | 116 |
| | 4 | D2 | AKBC | NORTH,COAST,SOUTH | 98 |
| | 4 | D3 | NORTH | AKBC,ALE,COAST | 86 |
| | 4 | D4 | NORTH | AKBC,COAST,SOUTH | 122 |
| | 4 | D5 | COAST | AKBC,NORTH,ALE | 92 |
| | 4 | D6 | COAST | AKBC,NORTH,SOUTH | 78 |
| | 4 | D7 | ALE | AKBC,NORTH,COAST | 110 |
| | 4 | **D8** | **SOUTH** | **AKBC,NORTH,COAST** | **136 (0.55)** |
| E | 1 | A2 | AKBC | | 46 |
| | 2 | B3 | AKBC | NORTH | 42 |
| | 3 | C2 | AKBC | NORTH,COAST | 127 |
| | 4 | D8 | SOUTH | AKBC,NORTH,COAST | 106 |
| | 5 | E1 | SOUTH | AKBC,NORTH,COAST,ALE | 133 |
| | 5 | E2 | AKBC | NORTH,COAST,SOUTH,ALE | 134 |
| | 5 | E3 | NORTH | AKBC,SOUTH,COAST,ALE | 120 |
| | 5 | **E4** | **ALE** | **AKBC,NORTH,SOUTH, COAST** | **151 (0.55)** |
| | 5 | E5 | COAST | AKBC,NORTH,SOUTH,ALE | 141 |

Stepwise comparison of demographic models of the invasion of *Mimulus guttatus* into the United Kingdom using 10,000 simulations for each of the model and random forest ABC model selection approach. At each step (model groups A–E), more complex introduction histories are considered while keeping the most likely models selected in previous comparison steps. The most likely model at each step is indicated in bold.

material to Russian, European and British collections. The colonisation of the UK by these exotic Aleutian monkeyflowers may have been facilitated by the close similarity of the ecological niche of *M. guttatus* in the British Isles and the Aleutian Islands[20]. Climatic pre-adaptation[44] of Aleutian monkeyflowers provided early arrivals with an opportunity for initial establishment. It is also clear that an initial introduction from the Aleutian Islands was accompanied or quickly followed by multiple introductions from other parts of the range. The UK seems to have become a melting pot for *M. guttatus* resulting in admixture of previously differentiated populations, which resulted in the creation of a unique set of genotypes that are now characteristic of UK populations (Fig. 4).

Invasive populations can themselves become sources for subsequent invasions, a phenomenon termed the "bridgehead effect"[12]. For example, the invasion of Australia by ragweed (*Ambrosia artemisiifolia*, Asteraceae) occurred not from native North American populations, but from populations in the introduced European range[23]. Our results indicate that UK populations served as a stepping-stone for secondary invasions in other parts of the non-native range. This potential bridgehead effect in invasive monkeyflowers is most clearly illustrated in the invasion of New Zealand. Some invasive populations in New Zealand share a close genetic affinity to UK populations. The genetic similarity is consistent with the exchange of biological material, including horticultural taxa, in the nineteenth century,

**Table 2 . (A) Power to discriminate between alternative demographic models using an "out-of-bag" procedure given the parameter model specification.**

**A.**

| Classified models Simulated models | A2 | B3 | C2 | D8 | E1 | E2 | E3 | E4 | E5 | Total | Classification error | Probability that E4 is selected |
|---|---|---|---|---|---|---|---|---|---|---|---|---|
| A2 | **8902** | 1043 | 24 | 12 | 2 | 9 | 0 | 1 | 7 | 10000 | 11.0% | 0.0% |
| B3 | 2151 | **7616** | 26 | 15 | 4 | 73 | 11 | 25 | 79 | 10000 | 23.8% | 0.3% |
| C2 | 210 | 463 | **4844** | 1576 | 675 | 642 | 534 | 469 | 587 | 10000 | 51.6% | 5.5% |
| D8 | 447 | 330 | 3067 | **1905** | 1039 | 662 | 908 | 911 | 731 | 10000 | 81.0% | 10.7% |
| E1 | 355 | 336 | 2094 | 1342 | **1462** | 735 | 1323 | 1339 | 1014 | 10000 | 85.4% | 15.8% |
| E2 | 400 | 1317 | 2161 | 1039 | 643 | **1473** | 756 | 1062 | 1149 | 10000 | 85.3% | 12.5% |
| E3 | 28 | 625 | 1894 | 1173 | 1277 | 857 | **1641** | 1438 | 1067 | 10000 | 83.6% | 17.0% |
| E4 | 344 | 1096 | 1024 | 993 | 1219 | 918 | 1313 | **2009** | 1084 | 10000 | 79.9% | **23.7%** |
| E5 | 435 | 1291 | 1491 | 777 | 963 | 1170 | 1006 | 1228 | **1639** | 10000 | 83.6% | 14.5% |
| Total | **13272** | **14117** | **16625** | **8832** | **7284** | **6539** | **7492** | **8482** | **7357** | | | |

The comparisons are made at the final selection step between the most likely one- to four-wave introduction models and all possible five-wave introduction models. The table shows how many of the 10,000 simulated datasets generated under a given scenario (A2 to E5, rows) were classified into each demographic scenario (A2 to E5 columns). The number of incorrect classifications is then used to compute the overall classification error. The last column shows the percentage of simulated models classified as E4 (which was the most likely scenario for the observed genetic dataset). Bold numbers indicate correct classification, and underlined numbers indicate >10% incorrect classification. (**B**) Probability of a given number of origins given that the E4 model is selected.

as British people migrated to New Zealand[45]. The single sampled population in continental Europe (Germany) also shows a close relationship to UK populations. Unfortunately, without further sampling it is difficult to establish whether UK populations contribute to the extant populations of *M. guttatus* in Europe. The hypothesis that UK populations have served as bridgeheads to other invasions remains to be further investigated. Morphologically, *M. guttatus* populations in Russia, Germany and the Czech Republic resemble UK material (Vallejo-Marín, *pers. obs.*) but the genetic identity of continental Europe populations remains to be investigated. In this regard, genomic analyses of herbarium specimens could provide important additional insights[46]. Particularly tantalising would be to compare specimens from herbaria in Russia, France and the UK, where historical links connect early *Mimulus* collections with Langsdorff's expedition to Alaska in the early 19th century. Finally, we also detected a close affinity between UK populations and a population in the non-native range in eastern North America. Populations of *M. guttatus* in eastern North America are generally small, occurring in the states of Michigan, New York, USA and in New Brunswick, Canada[27]. These small and sparsely distributed populations show diverse genetic origins and seem to be much more recently established (second half of the twentieth century). The mechanism of introduction of UK material into eastern North America is unknown but it could be associated with horticultural exchanges[1,5,47]. Our results combined with previous work on other systems (e.g., refs. [12,23]), highlight the importance that bridgehead populations can have on biological invasions. Bridgehead populations may have already been selected for their ability to colonise beyond their native range, making them particularly good candidates for additional range expansions.

Multiple introductions and admixture can, in principle, both increase or decrease the performance and adaptive potential of invasive populations[48–50]. Multiple introductions from genetically distinct sources introduce variation and alleviate the negative effects of demographic bottlenecks associated with colonisation. Moreover, genetically diverse populations are less likely to experience the deleterious effects of inbreeding depression[50,51] and can increase individual fitness through heterosis[49], which can potentially make invasive populations more difficult to control. In contrast, admixture may reduce overall fitness if gene flow results in outbreeding depression[52], a phenomenon that can occur due to epistatic interactions or, for

example, the breakdown of locally adapted genotypes. In *M. guttatus*, experimental work indicates that both positive and negative effects of admixture can be observed in invasive populations. For example, crossing native and introduced populations results in an increase in biomass, and both clonal and sexual reproduction in greenhouse conditions[53,54]. In field conditions, the effects of admixture can be reversed, and a common garden study shows that admixture between UK *M. guttatus* and both annual and perennial populations from the native range result in lower fitness as estimated using population growth rates[39]. The effects of admixture may be particularly strong on invasive species with a widespread, highly diverse native distribution, such as *M. guttatus*. Native populations that occur over large, biogeographically diverse areas may serve as reservoirs of genetic and ecological variation. This wide range of ecogeographic variation may facilitate the colonisation of new regions in the introduced range and potentiate the effects of subsequent introductions and admixture on the performance and adaptive potential of invasive populations.

## Methods

**Study system and population sampling**. *Mimulus guttatus* Fischer ex DC (section *Simiolus*, Phrymaceae), the common monkeyflower, is a widespread species with a native range extending across western North America from northern Mexico to the farthest reaches of the Aleutian Island chain in Alaska[20,33]. The invasive range includes much of the UK, the Faroe Islands, parts of mainland Europe, New Zealand, and Eastern North America[20]. The species is self-compatible and predominantly outcrossing[55]. Most populations are diploid, although tetraploid populations occur throughout the native range[56] and tetraploid populations have also evolved in the introduced range[35,56]. In the native range, populations comprise either small annual plants that reproduce exclusively by seed or perennial plants that reproduce by both seed and vegetative stolons. Only perennial plants are documented in the invasive range.

We sampled populations of *M. guttatus* in the native range of western North America and the main areas of introduction in eastern North America, Europe and New Zealand for a total of 521 individuals from 158 populations (Fig. 1 and Table 3). In the native range, the samples included 70 previously genotyped populations[34], spanning Arizona to British Columbia, plus an additional population from Vancouver Island. To fill the gap of previous studies, and to specifically address the hypothesis of an Alaskan origin of introduced UK populations, we collected samples from 32 populations in Alaska, including 14 populations from the Aleutian Islands (Attu, Unalaska, Akutan and Unimak) (Supplementary Data 1). Voucher specimens of the newly sampled populations are deposited in the University of Alaska herbarium (ALA). In the introduced range, we sampled four populations in eastern North America, one from the Faroe Islands, one from Germany, six from New Zealand, and 43 from UK populations from Cornwall to the Shetland Islands. As an outgroup we included three diploid individuals from a population of *M. glabratus* from Michigan, USA. We also

**Table 3 Number of populations and individuals sampled and sequenced.**

|  | Region | Number of populations | Number of individuals |
|---|---|---|---|
| Native | Western North America (excluding Alaska) | 71 | 182 |
|  | Western North America (Alaska only) | 32 | 106 |
| Introduced | Eastern North America | 4 | 34 |
|  | Faroe Islands | 1 | 4 |
|  | United Kingdom | 43 | 161 |
|  | Germany | 1 | 9 |
|  | New Zealand | 6 | 25 |
| Total |  | 158 | 521 |

Summary of the number of populations and individuals sampled and sequenced. A detailed breakdown by population is shown in Supplementary Data 1.

sampled three tetraploid UK *M. guttatus*, 19 individuals of *M. luteus* from both native and introduced ranges (with which *M. guttatus* hybridises in the introduced range to produce a sterile but widespread triploid, *M.* x *robertsii*), three *M.* × *robertsii*, and three *M. peregrinus* (the allohexaploid species derived by whole genome duplication from *M.* x *robertsii*[57]; (Supplementary Data 1). In total, we had samples from 103 populations of *M. guttatus* from the native range, and 55 populations from the introduced range (Table 3). Full sample details are provided in Supplementary Data 1.

**Genotyping**. To obtain DNA for genotyping, we germinated field-collected seeds from all new populations in a controlled environment facility at the University of Stirling. We extracted genomic DNA from fresh leaves or flower buds using the DNeasy Plant Kit (Qiagen, Germantown, MD), with samples standardised to 100 ng DNA for library preparation. We used genotyping by sequencing (GBS) to generate genome-wide polymorphism data[58]. For GBS library preparation, we used the same protocol as Twyford and Friedman[34], using the enzyme PstI, and pooling samples in a 95-plex (plus one blank water control) for 100 bp single-end sequencing on the Illumina HiSeq 4000 at the University of Oregon. We analysed raw sequence reads using the Tassel5-GBSv2Pipeline[59], using the *M. guttatus* v2 genome[60] as a reference. Alignment with the reference genome was done with BWA with default parameters in Tassel5-GBSv2Pipeline. To obtain variant sites we used a minimum allele frequency of mnMAF = 0.01. For the genotyping step (SNPCallerPluginV2) we used the following options: kmerLength = 64, mnQs = 20, minPosQS = 20. For population genomic analyses, we retained only variable sites (SNPs), but for estimating nucleotide diversity across the genome ($\pi_{GENOME}$) and for tree reconstruction, we generated a sequence matrix with both SNPs and invariant sites (setting mnMAF = 0).

To estimate pairwise nucleotide diversity across the genome of *M. guttatus* populations ($\pi_{GENOME}$), we filtered the data set containing both variant and invariant sites (123,141 loci) using VCFtools version 0.1.15[61]. We kept only biallelic loci that were genotyped in at least 75% of all individuals, which reduced the data set to 26,012 loci. We then removed individuals with <50% genotyped loci, reducing the number of individuals from 521 to 475. Nucleotide diversity per site was calculated separately for each of 155 populations, both native and invasive, using the option --site-pi in VCFtools. An overall pairwise nucleotide diversity across the ($\pi_{GENOME}$) for each population was then obtained by averaging across all sites.

For population genetic analyses in *M. guttatus*, we filtered the SNP data (44,552 loci from 521 *M. guttatus* individuals) using VCFtools version 0.1.15[61] and kept only biallelic loci that were genotyped in at least 75% of all individuals, which reduced the number of genotyped SNPs to 1820 loci. The parameters we used in this filtering step were: max-missing = 0.75, mac = 3, min-alleles = 1, max-alleles = 2. We then removed individuals with <50% genotyped loci, reducing the number of individuals from 521 to 474. Finally, we used PLINK version 1.9[62] to thin the data set to reduce linkage disequilibrium among SNPS using a pairwise correlation coefficient of 0.5 (--indep-pairwise 50 5 0.5). The final *M. guttatus* data set contained 1498 SNPs from 474 individuals in 155 populations (Supplementary Table 2).

**Tree building**. We sought to resolve evolutionary relationships between populations and species using polymorphism-aware phylogenetic models implemented in IQ-TREE[63]. These models use population site frequency data, and therefore account for incomplete lineage sorting[64]. This phylogeographic approach generates an initial visualisation of population history and broad scale geographic genetic structure from the genome-wide signal, prior to more detailed characterisation with population-level approaches (described below). We analysed two datasets, one for all sampled *Mimulus* taxa, and one for *M. guttatus*, with both datasets, including *M. glabratus* as an outgroup. Each analysis used the full GBS sequences with invariant sites (123,141 loci including both variant and invariant sites), filtered to include 8798 sites with <50% missing data. We calculated population allele frequencies using the counts file library (cflib) python scripts that accompany[64]. Model-fitting was performed with ModelFinder[65]. IQ-TREE analyses subsequently used the best-fitting model (TVM + F + G4) allowing for excess polymorphism

(+P) and with five chromosome sets per population (+N5). Tree searches were performed with settings recommended for short sequences, including a small perturbation strength (-pers 0.2) and large number of stop iterations (-nstop 500). Topological support was assessed using an ultrafast bootstrap approximation approach[66], with 1000 bootstrap replicates. Trees were visualised with *FigTree*[67].

**Population genetic structure**. To analyse population genetic structure, we conducted a principal component analysis using the *glPca* function in *adegenet*[68] in R ver. 4.0.0[69]. We used *K*-means grouping implemented with the function *find.clusters* in *adegenet* to identify clusters of individuals in the data without using a priori groupings. For this analysis, we used 100 randomly chosen centroids for each run, and calculated the goodness of fit for each model for values of *K* between two and 15. For the selected *K*-value, we also ran a discriminant analysis of principal components (DAPC)[70] using the inferred groups for assigning individual membership. We further used *fastStructure* version 1.0[71] to infer population structure across *M. guttatus* populations using a Bayesian framework. For this analysis, we randomly subsampled the data to include a maximum of three individuals per population (408 individuals in total) from both native and introduced ranges, and analysed values of *K* from 2 to 8.

**Introduction history reconstruction by ABC**. Our preliminary analyses indicated that introduced *M. guttatus* had a complex origin with multiple introductions in different non-native regions. In order to gain a more detailed understanding of the demographic history of non-native populations, we focused on the introduction of *M. guttatus* to the UK, which has been best studied historically and genetically[30,31]. Therefore, we implemented an approximate Bayesian computation (ABC) approach to determine the most likely *M. guttatus* introduction history in the UK. For this analysis, we used the pruned data set consisting of 1498 SNPs but included only individuals from the native range or the UK (399 individuals). Individuals from the native range were grouped into one of five groups ("genetic group"). Four of these groups (AKBC, NORTH, SOUTH and COAST) have been previously identified in analyses using native populations only[34,37]. The fifth group (Aleutian Islands, ALE) is identified in the present study (see Results section). To assign native individuals to each of these genetic groups, we used the group membership obtained in the *K*-means cluster analysis, which produced genetic groups that closely correspond to the native clades previously identified in Twyford and Friedman[34] and Twyford et al.[37] (see "Results" section). The number of individual in each group is as follows: NORTH, *N* = 62; SOUTH, *N* = 42; COAST, *N* = 30; AKBC (Alaska-British Columbia), *N* = 70; and ALE, *N* = 45. Six individuals from two populations (SWC and HAM) that formed a separate genetic group in the native range were not included in this analysis. Individuals from the UK were considered to belong to a single population (UK; *N* = 150).

As all possible scenarios of divergence between the five native groups would have been computationally impossible to test, native group genetic relationships were determined from the phylogenetic tree topology (see "Results" section). All the simulations assumed that the North population diverged from an ancestral population at time $t_4$, from which the South population diverged at time $t_5$. In addition, the Coastal population diverged from the ancestral population at time $t_3$ from which the Alaska-British Columbia population diverged at time $t_2$, and the Aleutian population diverged from there at time $t_1$. The simulated demographic models share this native population divergence history and only differed by their introduction history into the UK.

We first considered simple introduction models where the UK population was derived from a single native origin at time $t_{0a}$ (models A1 to A5, Supplementary Notes 1). We then simulated UK introduction from a single origin at time $t_{0a}$ followed by a second introduction at time $t_{0b}$ (two-waves introduction models; models B). This strategy resulted in the definition of eight different two-waves introduction models (models B1 to B8, Supplementary Notes 1). We then tested more complex introduction models using a similar logic, modelling three-waves (models C1 to C9), four-waves (models D1 to D8) and five-waves (models E1 to E5) introduction models by integrating the most likely origins identified in

previous sets of models to define a restricted number of models to compare. A full version of other assumptions and simulation parameters is given in Supplementary Notes 1.

For each demographic model, we simulated 10,000 genetic datasets consisting of 1435 independent SNP genotypes for 798 haploid individuals distributed following the sample size of all six populations in the real data set using *Fastsimcoal2* version 2.6.0.3[72] called by *ABCtoolbox* version 1[73]. We passed a custom bash script to *ABCtoolbox* to add missing genotypes to the simulated data set at an identical rate to the observed level in the real data. Then, we used *ABCtoolbox* to call the *arlsumstat* programme[74] to compute summary statistics from the simulated genotypes that were combined into diploid genotypes by randomly pairing haploid genotypes assuming Hardy-Weinberg equilibrium[75]. We computed all available statistics within and between populations for biallelic loci (67 summary statistics). In addition, we computed summary statistics within and between three defined regional groups (NORTH and SOUTH in one group; COASTAL, AKBC and ALE in a second group; and UK in a third group) representing an additional set of 29 summary statistics.

**ABC model comparisons.** We performed iterative model comparisons by comparing increasingly complex models (Table 1). In the first round, the introduction models assume a single introduction from one of the five native genetic groups. Then in round two, we considered two introductions models that necessarily involved the population origin from round one. This allowed us to define two sets of two-waves introduction models: one set consisting of four models with the most likely origin in previous rounds as the first introduction origin, followed by a second introduction from one of the four other native populations. Moreover, a second set of four models, which assume that the most likely origin in the previous round constitutes the second introduction, while the first introduction originated from one of the four other native populations (Table 1). We compared the most likely single introduction model and the eight two-waves introduction models. We then considered more complex models, comparing nine three-waves introduction models and the most likely single and two-waves introduction models (Table 1). We subsequently compared models assuming four-waves and five-waves of introduction while still including more simple models in the comparisons (Table 1). Demographic models were compared using a random forest approach implemented in the *R* package *abcrf*[76].

We built a classification random forest model using 1000 trees and a training data set consisting of the summary statistics computed for the 10,000 simulated genetic datasets for each model. We estimated the classification error rate for each model using an "out-of-bag" procedure to quantify the power of the genetic data given the models and prior distribution specifications to differentiate the different demographic models. Then, we used the summary statistics computed based on the observed genotypic data to predict the demographic model that best fit the data using a regression forest with 1000 trees. We report the number of "votes" for each demographic scenario and the approximation of the posterior probability of the most likely model. We used the overall most likely scenario to simulate 100,000 genetic datasets using parameters and prior distributions described above to estimate demographic model parameters. We built a regression random forest model implemented in *abcrf* based on the summary statistics using 1000 trees. We estimated the posterior median, 0.05 and 0.95 quantiles of the model parameters by random forest regression model based on the summary statistics of the observed genotypic composition.

**Reporting summary.** Further information on research design is available in the Nature Research Reporting Summary linked to this article.

## Data availability

Genotype data as a VCF file is publicly available at http://hdl.handle.net/11667/168 (DATAStorre, U. Stirling). Location data of sampled populations is available in the Supplementary Materials (Supplementary Data 1). Source data of Figs. 3 and 4 are provided in Supplementary Data 2. Herbarium specimens of newly collected material in Alaska is deposited at the ALA herbarium.

## Code availability

Code for the Approximate Bayesian Computation Analysis is available at https://doi.org/10.15454/VUMC1P.

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

## Acknowledgements

We thank John Willis and current and former members of his lab, including David Lowry and Kevin Wright, for providing access to North American seed material collected over many years, and to the Botanical Society of Britain and Ireland for their continued support locating UK *Mimulus* populations. Arielle Cooley kindly provided seed material from Chilean populations of *M. luteus var. variegatus*. We are very grateful to Claudia Buser and John Bailey for providing the New Zealand material, and Nils Bunnefeld, Anna Maria Fosaa and Símun Arge for their help while collecting *Mimulus* in the Faroe Islands. We thank Oregon Genomics (University of Oregon) for sequencing services, the University of Stirling Controlled Environment Facility for access to plant growth facilities, and Sophie Webster for help in the laboratory. Computer time for the ABC analysis was provided by the computing facilities MCIA (Mésocentre de Calcul Intensif Aquitain) of the Université de Bordeaux and of the Université de Pau et des Pays de l'Adour. L.Y. was supported by the European Research Council (ERC) under the European Union's Horizon 2020 research and innovation programme [grant number ERC-StG 679056 HOTSPOT], via a grant to L.Y. This work was supported by NSF 1754080 to J.R.P. This project was made possible by a grant from the Global Exploration Fund, Northern Europe from National Geographic (GEFNE164-15) to M.V.M., J.R.P. and S.M.I.-B., and a grant from the Natural Environment Research Council (NERC; NE/J012645/1) to M.V.M. We thank all the people who helped us during fieldwork in Alaska, particularly Suzi Golodoff (Unalaska Island) and Stacy Studebaker (Kodiak Island) for providing their exceptional knowledge of the local flora, and Roger Topp (U. Alaska, Fairbanks/Museum of the North), who documented the expedition with his outstanding photographs and video.

## Author contributions

M.V.M., J.R.P., S.M.I.B., J.F. and A.D.T. designed the research. M.V.M., J.R.P., J.F., S.M.I.B., M.C.R. and M.v.K. collected material. M.V.M., A.D.T., and O.L. analysed the data. M.V.M., J.F., L.Y., A.D.T. and J.R.P. wrote the manuscript with input from all the authors.

## Competing interests

The authors declare no competing interests.
