## [Peer Review File · Communications Biology]

Reviewers' comments:

Reviewer #2 (Remarks to the Author):

Review

In this paper, using historical records and genetic data, Vallejo-Marin et al. try to reconstruct the recent (19th century) demographic history of monkeyflowers (*M. guttatus*) in the United Kingdom. The (putative) native range of this species is located in the United States, and, in contrast to previous studies, samples from Aleutian islands are here included, completing the global overview and potential increasing the power of statistical analyses.

The major claim of this article is that, using ABC and random forest classification, the most likely demographic scenario is a multiple introduction scenario (from US to UK), and that UK has served as a bridgehead to further invasion (Europe and New Zealand). This paper is a technical demonstration that inferring recent demographic history can highly benefit from priors coming from historical records. Moreover, studying species that successful is a key feature to understand evolutionary mechanisms underlying invasiveness. Finally, this paper shows that integrating data consistent with the species distribution (worldwide scale) is essential to reconstruct a comprehensive evolutionary history of a species. It is with no doubt of interest for the community and the readers of this journal, and the methods used here are solid and convincing.

I only have some minor comments on the statistical work.

- in the native range, the authors propose five genetic groups in I.222 (ALE, NORTH, SOUTH, COAST and AKBC), that I do not fully understand how it was deduced... Is it from a preliminary work? Or is it from the k-mean clustering in figure 4, with the introduced population (UK, Europe and New Zealand)?

In the latter case, I suggest removing every population other than the native range population to do the clustering, to avoid sampling issues, especially considering that sampling in UK is overwhelming and might be biasing this process. Finally, can we have a PCA (like figure 3) colored only with the five groups?

- to continue on the sampling issues, and a more naive question, is it possible that the unequal sampling from UK and other places creates some biases : the UK is much more represented than the other sampling sites. In the tree building, the authors perform 1000 bootstrap with IQ-TREE (I.189). Is it sufficient to avoid sampling issues? Otherwise, the fact that New Zealand is branched with UK samples can come from an artefact : the sample best represent the diversity in UK, not in NZ because of sampling issues, thereby clustering these together per (bad) luck.

- "unexpected placement of from inland oregon" (I.297), do we have any explanation on that? Is it an artefact, in which case the methodology might be reconsidered, or is it a particularity of this population?

- the authors perform a cross validation summarized in Table 3. It seems that classification error is huge, thus concluding from this method seems rather challenging. Also, I do not understand where the 83.4% (from I.365) does come from. It appears from Table 3 that, when simulating the model E4, ABC has much trouble distinguishing any model, not only between five-waves models. A potential other issue is that the number of five-waves model is higher than the number less-waves model, potentially biasing the interpretation: it appears to me here that ABC is randomly assigning the model (and just slightly more to E4), and that when averaging the number of simulations per number of waves, the five-waves models have only 1308 simulations which is only slightly more than the others. I will be 1) interested to know the same Table with the same amount of model per number of waves, and 2) interested from the authors to discuss about this apparently high classification error (and if it is not that high, it will be interesting to have a classic reference value from other articles).

- I understood that historical records corroborate the history proposed by the authors, but is it not possible that admixture in the United States occurred prior to introduction in the UK? Maybe admixed population have been introduced, alternatively explaining the apparent multiple introduction into one

or perhaps two. I will be interested to have a further discussion about how we can exclude (or not) this hypothesis, with the historical records but also with the genetic data.

- I would nuance the bridgehead section, the authors have indeed inferred a introduction story in the UK, but have not shown per se any bridgehead effect: it would have needed, for instance in Europe, an ABC analysis opposing bridgehead vs direct introduction. It is, as the authors said, for further research that needs further sampling. I agree that the tree in Figure 2 corroborates this conclusion, but maybe a bit more caution would be harmless.

Typo :

- is it possible to specify the proportion of variance explained by each PCA axis? Knowing the relative importance of axes might help the interpretation (dealing with lots of SNPs fatally leads to small values which is not a problem).

- l.73 : "..., but beyond this little is known".

Reviewer #3 (Remarks to the Author):

In this manuscript, Vallejo-Marín et al. use numerous individuals of *Mimulus guttatus* species collected across the actual distribution range to investigate the demographic history of this species. They used genomic data (GBS, ~1500 informative SNPs) to reconstruct the recent history of successive invasion. Using a Bayesian approach, they confirm an initial introduction of *M. guttatus* in UK from Aleutian Islands, which was supported by historical records. In addition, the authors also point out that the invasion in other areas, up to New Zealand, was achieved through a bridgehead effect from the UK.

Globally, the paper is well written and the main findings clearly exposed and well supported by the analyses, I reckon that certain information are missing, in particular in the material and methods to allow a complete assessment of the relevance and correctness of the analyses conducted. I also have several questions / comments / suggestions on different aspects that could be improve. Please consider the following comments as a way to make the manuscript easier to understand for the reader.

Majors comments:

1. "genotyping" section. There is no precision about the parameters that the authors used for read mapping, SNP calling and further filtering. They used Tassel5-GBSv2pipeline but with which set of parameters? the default parameters? E.g. were the reads mapped against the reference genome using BWA or bowtie2? I strongly encourage the authors to developed this part in specifying all the parameters used. E.g.: What about filter base on coverage? Did any filtering was done after SNP calling (e.g. excess of heterozygote at HWE suggesting read collapsing)?

2. For population structure analyses, the authors indicated that they only keep bi-allelic loci that were genotyped in at least 75% of the individuals. What about minor allele frequency? Where singletons conserved? Did they filtered out SNPs below a certain MAF? is the same set of SNPs that used for PCA, faststructure, and approaches based on SFS?

3. It is normally recommended to carry out population structure analyses, especially when interested in demography history, on putatively neutral SNPs. Are the 1,498 SNPs all neutral or putatively neutral? Did the authors annotated the SNPs dataset using the available gene models for *Mimulus guttatus*? If not, please justify why.

4. ABC analysis: the authors focused only on individuals from the native range and the UK. One would expect such a down-sampling to generate invariant sites (especially if no filtering for low MAF was applied). However, the number of SNPs seems to be the same than that used in the previous analyses with the full dataset. Could the authors clearly report the number of samples and the number of SNPs used in each analysis, maybe in a table.

5. Finally, I think that it could be of interest for the reader to add a part with more commonly used estimates in population genetics. It would have the merit to help the readers that are not specialist of demography history modelling and therefore to reach a broader audience. For example, pairwise F_{ST} (Weir and Cockerham's given the type of data) between the different main clusters defined using the k-means approach would provide a direct reading of the genetic differentiation between the different clusters. In the same line, nucleotide diversity (π) could be also reported for the different clusters. In particular, it could be then related to the demography history of the species. In a very simplistic view, one would expect π to be higher in the in native populations than in the recently invaded ones, except in areas of multiple introduction.

6. Why simulating haploid individuals? How does this could affect the estimates of the model? I think that this deserve to be discussed in depth, especially knowing that different level of ploidy is encountered in the species.

Minor comments:

- Line 74: Please define the bridgehead process. (as for example- "many invasions could be due to a particularly successful invasive populations and not due to individuals from the native range). Perhaps add an example of invasion via bridgehead.

- Line 115: The niche shift was only characterised based on climatic data? Please, provide details of the parameters used to define the niche.

Are the niche also similar in terms of species composition or soil composition?

- Line 391: Could you give an example of a phenotypic trait that have changed between native and admixed populations in the introduced range?

- Line 440: There are surely many examples, from plant or animal models, for which the same climatic conditions can help the initial establishment. Could you cite another one to validate this key point of the invasion?

- Line 481: Could you explain and develop this idea. What are the potential consequences of multiple origins at different times?

For instance, it increases genetic diversity and I suppose that it could be a problem in terms of control of invasive species.

- Line 473: I think it would be interesting to discuss one other point in this part. Individuals that first colonized and got established in the UK are, by definition, apt to colonize (maybe better competitors than other genotypes or so). This could also partly explain why this cluster (UK) was then the source of new invasions in several parts of the world (Europe, New Brunswick, New Zealand and Germany). Do ecological or life history traits data exist in the literature that would support that hypothesis?

- Line 36, in supplementary data. 'All native populations and 36 the ancestral population were assumed to have constant effective population size'. Why did the authors make such an assumption?

Are some data exist that would suggest such a thing? Did they test models with different N_e ? This should be at least discussed.

- Please add, for the different programs/pipelines, the version you are using
- Figure 1. Please divide the figure into panels (e.g. A to D) to ease reading.
- Figure 2. I like the interactive plot with the three first dimension and I am wondering if projection methods as UMAP (Uniform Manifold Approximation and Projection) could be used in order to improve the visualization of the clustering in the main text. It could then replace the two PCA plots by a single figure containing the same information.
- Figure 4. The capital letters for each panel are missing
- Figure S2. It would be interesting to provide a plot for "thermotical" best K definition. Is it also $K=8$, as for clustering based on PCA? It is an important point as I suppose that the five groups used for ABC reconstruction were derived from this analysis.

Response to reviewers

Below we provide our response to the Reviewers. Their original comments are shown in italics, and our response is provided immediately after in blue font. All the new text in the revised version of the manuscript is highlighted in yellow.

Reviewer #1

*In this paper, using historical records and genetic data, Vallejo-Marin et al. try to reconstruct the recent (19th century) demographic history of monkeyflowers (*M. guttatus*) in the United Kingdom. The (putative) native range of this species is located in the United States, and, in contrast to previous studies, samples from Aleutian islands are here included, completing the global overview and potential increasing the power of statistical analyses. The major claim of this article is that, using ABC and random forest classification, the most likely demographic scenario is a multiple introduction scenario (from US to UK), and that UK has served as a bridgehead to further invasion (Europe and New Zealand). This paper is a technical demonstration that inferring recent demographic history can highly benefit from priors coming from historical records. Moreover, studying species that successful is a key feature to understand evolutionary mechanisms underlying invasiveness. Finally, this paper shows that integrating data consistent with the species distribution (worldwide scale) is essential to reconstruct a comprehensive evolutionary history of a species. It is with no doubt of interest for the community and the readers of this journal, and the methods used here are solid and convincing. I only have some minor comments on the statistical work.*

We sincerely thank the referee for this positive and encouraging assessment of our manuscript.

- 1) *in the native range, the authors propose five genetic groups in I.222 (ALE, NORTH, SOUTH, COAST and AKBC), that I do not fully understand how it was deduced... Is it from a preliminary work? Or is it from the k-mean clustering in figure 4, with the introduced population (UK, Europe and New Zealand)? In the latter case, I suggest removing every population other than the native range population to do the clustering, to avoid sampling issues, especially considering that sampling in UK is overwhelming and might be biasing this process.*

Population assignment to different genetic groups in the native range was guided by previous work (Twyford and Friedman, 2015) that had identified 4 of the 5 groups we used (NORTH, SOUTH, COAST, AKBC) using exclusively individuals in the native range as suggested by the reviewer. The fifth group (ALE) represents the newly sampled Aleutian populations and it was deduced from both k-means analysis, ML tree reconstruction and *fastStructure* analyses. In our study we used k-means to delimit genetic group membership. For native-range samples, the groups identified largely corresponded to one of the previously identified four groups (NORTH, SOUTH, COAST, AKBC) plus ALE. These group memberships were used in the ABC analysis. Reassuringly, our grouping results recover the same groupings that were obtained with only native individuals by Twyford and Friedman (2015) and Twyford et al. (2020). Thus, the groups used here for native range samples are robust and consistent with previous work based on native populations only. We have modified the text in lines 227-236 to clarify this point.

- 2) *Finally, can we have a PCA (like figure 3) colored only with the five groups?*

As requested, the five groups (ALE, AKBC, NORTH, SOUTH, COAST) for native populations mentioned by the reviewer have now been clearly coloured and labelled in the PCA shown in Figure 4C. We

have added a colour polygon to facilitate the visual recognition of the groups and also added a text annotation with the corresponding labels (ALE, AKBC, NORTH, SOUTH, COAST).

- 3) *to continue on the sampling issues, and a more naive question, is it possible that the unequal sampling from UK and other places creates some biases : the UK is much more represented than the other sampling sites. In the tree building, the authors perform 1000 bootstrap with IQ-TREE (l.189). Is it sufficient to avoid sampling issues? Otherwise, the fact that New Zealand is branched with UK samples can come from an artefact : the sample best represent the diversity in UK, not in NZ because of sampling issues, thereby clustering these together per (bad) luck.*

We thank the reviewer for these insightful questions. The 1000 bootstraps should be sufficient to assess the robustness of the inferred topology. Importantly, a large UK sample would not be sufficient to explain the clustering of some of the New Zealand samples within the UK as demonstrated by the fact that other New Zealand samples do not cluster in the UK clade in the ML tree but instead cluster with the COAST clade (Figure 2). In fact, from the 6 populations from New Zealand analysed, three cluster in the UK clade and three in the COAST clade. This result is robust to the analytical method chosen as demonstrated by similar conclusions derived from the k-means and fastStructure analyses.

- 4) *“unexpected placement of from inland oregon” (l.297), do we have any explanation on that? Is it an artefact, in which case the methodology might be reconsidered, or is it a particularity of this population?*

This is an excellent question and one for which we do not have a satisfactory answer. The unusual placement of the clade of “inland Oregon” in the ML tree is puzzling. Two of the populations in this clade (HAM and SWC) are identified as being genetically distinct from other native groups in all analyses in addition to the ML tree (eg PCA analysis). However, for the remaining native populations in this clade, the PCA analysis shows more affinity with populations in the NORTH clade. We believe that this distinctiveness in the ML tree may be partly real in the sense that extensive gene flow or genetic differentiation in these populations makes phylogenetic inference more difficult, and partly methodological as the ML analyses assume bifurcating relationships. In this regard, we think that the PCA, k-means and fastStructure analyses may better reflect the genetic relationship of these particular populations.

- 5) *the authors perform a cross validation summarized in Table 3. It seems that classification error is huge, thus concluding from this method seems rather challenging. Also, I do not understand where the 83.4% (from l.365) does come from. It appears from Table 3 that, when simulating the model E4, ABC has much trouble distinguishing any model, not only between five-waves models. A potential other issue is that the number of five-waves model is higher than the number less-waves model, potentially biasing the interpretation: it appears to me here that ABC is randomly assigning the model (and just slightly more to E4), and that when averaging the number of simulations per number of waves, the five-waves models have only 1308 simulations which is only slightly more than the others. I will be 1) interested to know the same Table with the same amount of model per number of waves, and 2) interested from the authors to discuss about this apparently high classification error (and if it is not that high, it will be interesting to have a classic reference value from other articles).*

We updated the cross-validation analysis by only considering the best one- to five-waves introductions models and comparing 10,000 simulations for each (Table 3). High classification error is likely due to the wide prior distribution we used for migration rate, so that if an origin was

introduced with a very small number of individuals relative to the population size in UK (as low as 0.1% of the UK effective population size), this introduction event will be most probably undetected. This will lead to increased noise when differencing between scenario with high number of introductions. We added some clarification about this on lines 416-427, and found that the moderate level of confidence in scenario choice that we observed is typical of complex scenario with multiple introductions (Framout et al. 2017. *Molecular Biology and Evolution*. 34(4):980-996. <https://doi.org/10.1093/molbev/msx050>).

- 6) *I understood that historical records corroborate the history proposed by the authors, but is it not possible that admixture in the United States occurred prior to introduction in the UK? Maybe admixed population have been introduced, alternatively explaining the apparent multiple introduction into one or perhaps two. I will be interested to have a further discussion about how we can exclude (or not) this hypothesis, with the historical records but also with the genetic data.*

The possibility of admixture occurring in the native range in North America before *M. guttatus* was introduced into the UK is certainly possible. However, we find no evidence to support this possibility. Admixture in the native range would be detectable in our analysis, for example by showing a distinct admixed group in samples collected in the native range. However, this is not the case in any of the different analyses we conducted (trees, k-means and fastStructure). It is possible that a putatively admixed native source population has since become extinct and therefore not sampled in our study. However, together with the historical evidence of single-source initial introduction (Aleutians) we believe that our explanation following introduction to the UK is more likely.

- 7) *I would nuance the bridgehead section, the authors have indeed inferred a introduction story in the UK, but have not shown per se any bridgehead effect: it would have needed, for instance in Europe, an ABC analysis opposing bridgehead vs direct introduction. It is, as the authors said, for further research that needs further sampling. I agree that the tree in Figure 2 corroborates this conclusion, but maybe a bit more caution would be harmless.*

We agree with the reviewer and we previously tried to be cautious with our inferences in the bridgehead role of the UK on the invasion to other parts of Europe (Germany) as shown in line 457: "Unfortunately, without further sampling it is difficult to establish whether UK populations contribute to the extant populations of *M. guttatus* in Europe." In addition, we now have further modified the Discussion to add a bit more caution in our interpretation and suggest that more work is needed in this regard: line 458-460: "The hypothesis that UK populations have served as bridgeheads to other invasions remains to be further investigated."

- 8) *is it possible to specify the proportion of variance explained by each PCA axis? Knowing the relative importance of axes might help the interpretation (dealing with lots of SNPs fatally leads to small values which is not a problem).*

We are glad to follow the suggestion by the reviewer. We have now incorporated the percent of variance explained by each of the three principal components to (PC1: 8.39%, PC2: 7.2%, PC3:4.01%) to the axis labels of Figures 3 and 4.

- 9) *1.73 : "..., but beyond this little is known".*

Typo corrected.

Reviewer #2

In this manuscript, Vallejo-Marin et al. use numerous individuals of Mimulus guttatus species collected across the actual distribution range to investigate the demographic history of this species. They used genomic data (GBS, ~1500 informative SNPs) to reconstruct the recent history of successive invasion. Using a Bayesian approach, they confirm an initial introduction of M. guttatus in UK from Aleutian Islands, which was supported by historical records. In addition, the authors also point out that the invasion in other areas, up to New Zealand, was achieved through a bridgehead effect from the UK.

Globally, the paper is well written and the main findings clearly exposed and well supported by the analyses, I reckon that certain information are missing, in particular in the material and methods to allow a complete assessment of the relevance and correctness of the analyses conducted. I also have several questions / comments / suggestions on different aspects that could be improve. Please consider the following comments as a way to make the manuscript easier to understand for the reader.

- 10) *"genotyping" section. There is no precision about the parameters that the authors used for read mapping, SNP calling and further filtering. They used Tassel5-GBSv2pipeline but with which set of parameters? the default parameters? E.g. were the reads mapped against the reference genome using BWA or bowtie2? I strongly encourage the authors to developed this part in specifying all the parameters used. E.g.: What about filter base on coverage? Did any filtering was done after SNP calling (e.g. excess of heterozygote at HWE suggesting read collapsing)? For population structure analyses, the authors indicated that they only keep bi-allelic loci that were genotyped in at least 75% of the individuals. What about minor allele frequency? Where singletons conserved? Did they filtered out SNPs below a certain MAF? is the same set of SNPs that used for PCA, faststructure, and approaches based on SFS?*

We thank the referee for this suggestion. We have now added details about specific parameters used and the aligning algorithm (BWA) to the Methods section (lines 168-173). Briefly, alignment with the reference genome was done with BWA with default parameters. For variant sites we used a minimum minor allele frequency $mnMAF = 0.01$. For the genotyping step (SNP Caller Plugin V2) we used the following options: $kmerLength = 64$, $mnQs = 20$, $minPosQS = 20$. Furthermore, we have re-organised the presentation to make clearer how the additional filtering after SNP calling (genotyping) was conducted for population genomic analyses. The filtering steps are described in lines 174-180. To this text we have also added the specific parameters used in VCFTools: $max-missing = 0.75$, $mac = 3$, $min-alleles = 1$, $max-alleles = 2$. Finally, the reviewer is correct that the exact same data set was used for all population genomic analyses. The data set used for tree reconstruction was different (consisting of 8,798 loci including both variant and invariant sites) and keeping only sites with less than 50% missing data (lines 193-194).

- 11) *It is normally recommended to carry out population structure analyses, especially when interested in demography history, on putatively neutral SNPs. Are the 1,498 SNPs all neutral or putatively neutral? Did the authors annotated the SNPs dataset using the available gene models for Mimulus guttatus? If not, please justify why.*

In a pilot analysis we did try to separately analyse loci within the known chromosomal inversions (cf. Twyford and Friedman 2015), but unfortunately the small number of loci contained in the inversions in our final data set (48 loci) was too small to conduct meaningful analyses. We did not annotate the 1,498 SNPs using gene models. Our approach of including all 1,498 loci was based on trying to provide a first approximation to the genetic relationships between native and introduced species using the broadest coverage of the genome possible with our data set. We believe that our results

are not strongly influenced by non-neutrality because the fraction of loci under strong selection is likely to be small in this genome-wide random sampling of the monkeyflower genome which includes only ~1500 loci. However, future work using genotyping methods allowing to interrogate the genome at much higher depth (e.g., whole genome sequencing) could be employed in the future to assess the extent to which demography and selection have shaped the population genomic history of introduced populations. We are currently pursuing this line of research using WGS on selected populations and hope to be able to provide further insights in this regard in the near future.

12) ABC analysis: the authors focused only on individuals from the native range and the UK. One would expect such a down-sampling to generate invariant sites (especially if no filtering for low MAF was applied). However, the number of SNPs seems to be the same than that used in the previous analyses with the full dataset. Could the authors clearly report the number of samples and the number of SNPs used in each analysis, maybe in a table.

We hope that the edits that we have made in the Methods now provide further clarity that in fact we did use stringent filtering steps, including minimum allele frequencies (MAF). We also recognise that the number of sites and individuals analysed was difficult to find in the manuscript as it was provided in the text in different sections. To improve clarity, we now provide a Table in the supplementary material (Table S3) with the number of loci and individuals used in each of the analysis. This should help the reader quickly assess sample sizes for each of the analyses conducted. We thank the reviewer for their suggestion.

13) Finally, I think that it could be of interest for the reader to add a part with more commonly used estimates in population genetics. It would have the merit to help the readers that are not specialist of demography history modelling and therefore to reach a broader audience. For example, pairwise F_{ST} (Weir and Cockerham's given the type of data) between the different main clusters defined using the k-means approach would provide a direct reading of the genetic differentiation between the different clusters. In the same line, nucleotide diversity (π) could be also reported for the different clusters. In particular, it could be then related to the demography history of the species. In a very simplistic view, one would expect π to be higher in the in native populations than in the recently invaded ones, except in areas of multiple introduction.

We thank the referee for this suggestion. We have calculated genome-wide nucleotide diversity (π_{genome}) to provide interested readers with a measure of genetic diversity that is commonly used in genomic studies. We computed this measure for each of 155 populations and present a comparison between regions (both native and introduced) in Figure SX. This measure of diversity shows that UK populations indicate a relatively high level of nucleotide diversity on average as expected from its admixed history and correctly guessed by the reviewer. However, we urge caution with the interpretation of broad comparisons among regions or groups as there is a significant amount of variation in genetic diversity among populations within the same region (Figure SX), and we have a limited number of populations in some regions (eastern North America, New Zealand, Faroes, Germany; Figure S5). We have edited the text in lines 175-176 and 364-372 to reflect this new analysis and report its findings.

14) Why simulating haploid individuals? How does this could affect the estimates of the model? I think that this deserve to be discussed in depth, especially knowing that different level of ploidy is encountered in the species.

Coalescent simulation works by simulating haplotypic information within populations, and thus haploid genotypes at the marker level. Because only diploid genotypes were observed in real

populations, and as we computed some summary statistics that were only relevant for diploid genotypes (for instance, level of heterozygosity), it is usual practice to randomly combine haploid genotypes to form diploid genotypes assuming Hardy-Weinberg equilibrium (as explained for example in Ray et al. 2010. *Bioinformatics*. 26(23):2993-2994. doi:10.1093/bioinformatics/btq579). We added an explanatory note on lines 286-287.

15) *Line 74: Please define the bridgehead process. (as for example “many invasions could be due to a particularly successful invasive populations and not due to individuals from the native range). Perhaps add an example of invasion via bridgehead.*

Definition added and also included two references of examples of invasion via bridgehead in animals and plants.

16) *Line 115: The niche shift was only characterised based on climatic data? Please, provide details of the parameters used to define the niche.*

We have edited this to “climatic niche” to emphasise that the niche shift study was based on climatic data.

17) *Are the niche also similar in terms of species composition or soil composition?*

Currently no study has compared the species or soil composition of native and introduced populations of *M. guttatus*.

18) *Line 391: Could you give an example of a phenotypic trait that have changed between native and admixed populations in the introduced range?*

Example and citation added as suggested to exemplify how invasive phenotypes include the combination of high levels of clonality and seed fertility.

19) *Line 440: There are surely many examples, from plant or animal models, for which the same climatic conditions can help the initial establishment. Could you cite another one to validate this key point of the invasion?*

Thanks for this suggestion. We have added a reference to strengthen this point.

20) *Line 481: Could you explain and develop this idea. What are the potential consequences of multiple origins at different times? For instance, it increases genetic diversity and I suppose that it could be a problem in terms of control of invasive species.*

Repeated introduction events will potentially increase admixture with both benefits through increase genetic variation among introduced populations and disadvantages due to outbreeding depression. This argument is discussed in the following sentences and the remaining of the mentioned paragraph. We have edited the text to explicitly mention that the positive effects of admixture could make invasive populations more difficult to control (line 490).

21) *Line 473: I think it would be interesting to discuss one other point in this part. Individuals that first colonized and got established in the UK are, by definition, apt to colonize (maybe better competitors than other genotypes or so). This could also partly explain why this cluster (UK) was then the source of new invasions in several parts of the world (Europe, New Brunswick,*

New Zealand and Germany). Do ecological or life history traits data exist in the literature that would support that hypothesis?

This is a very interesting hypothesis, and we are very thankful to the reviewer for suggesting it. We now discuss this in lines 482-486. Future studies could put this hypothesis further to the test, although a full analysis of the strengths and weaknesses of this hypothesis and whether is supported by other studies is beyond the scope of the present study.

22) Line 36, in supplementary data. 'All native populations and 36 the ancestral population were assumed to have constant effective population size'. Why did the authors make such an assumption? Are some data exist that would suggest such a thing? Did they test models with different Ne? This should be at least discussed.

We assumed constant effective population sizes for native and ancestral population to be parsimonious with the number of parameters used to build the demographic models. Even with this simplification, the demographic models are already too complex to estimate the demographic parameters using the data at hand. Furthermore, we have no clear hypothesis about historical effective population sizes variation through time in native or ancestral populations. As a result, parametrizing variable effective population sizes would only add more parameters and more noise to an already complex problem. Two different timescales are involved here: (i) long history of native population diverging from each other and from ancestral populations, versus (ii) short time scales since the introduction of *Mimulus* to UK. In this situation, long-term effective population size variation is best averaged over a single fixed effective population size (that is not a parameter of direct interest). On the contrary it is important to account for very recent population dynamics in the UK because quick population expansion should leave some specific genetic diversity pattern in the introduction range. We now added an explanation in the supplementary data.

23) Please add, for the different programs/pipelines, the version you are using

Done.

24) Figure 1. Please divide the figure into panels (e.g. A to D) to ease reading.

Done.

25) Figure [3]. I like the interactive plot with the three first dimension and I am wondering if projection methods as UMAP (Uniform Manifold Approximation and Projection) could be used in order to improve the visualization of the clustering in the main text. It could then replace the two PCA plots by a single figure containing the same information.

Thank you very much for this suggestion. We conducted a visualisation using UMAP to summarise multidimensional information (McInnes, Healy & Melville 2018). This method allowed us to summarise the information contained in the PCA scores into a reduced number of dimensions (2 dimensions) as suggested by the reviewer. We used the first 200 principal components, and the results are qualitatively similar to those captured in the first three principal components shown in Figure 2. We think that showing the results in a PCA is still a better option to present in the main text as it is an analysis that is commonly used in similar studies and one that many researchers are familiar with. In addition, in UMAP the axes (V1 and V2) do not have straightforward interpretations as in the case of PCA, and the distance between groups of points is not proportional to their differentiation (McInnes et al 2018). We would like to respectfully request to keep Figure 3 as is, and

instead provide the new UPMA visualisation as a supplementary figure for the interested reader. We have added Figure S4 to depict the UPMA visualisation of the first 200 principal components.

26) *Figure 4. The capital letters for each panel are missing*

The capital letters have now been added

27) *Figure S2. It would be interesting to provide a plot for “thermotical” best K definition. Is it also K=8, as for clustering based on PCA? It is an important point as I suppose that the five groups used for ABC reconstruction were derived from this analysis.*

The groupings for ABC reconstruction were derived from the clustering method shown in Figure 4 (for additional detail see response to Reviewer 1), and this is why we chose to present the K-estimation in more detail in Figure 4C. Following the reviewer’s suggestion, we estimated the “optimal” K using two methods developed for fastStructure analyses (Puechmaile SJ (2016) The program structure does not reliably recover the correct population structure when sampling is uneven: subsampling and new estimators alleviate the problem. *Molecular Ecology Resources*, 16:608–627; and Raj A, Stephens M, Pritchard JK (2014) fastStructure: Variational Inference of Population Structure in Large SNP Data Sets. *Genetics*, 197:573-589., implemented in Li, Y.-L., & Liu, J.-X. (2018). StructureSelector : A web-based software to select and visualize the optimal number of clusters using multiple methods. *Molecular Ecology Resources*, 18(1), 176-177. doi:10.1111/1755-0998.12719). Both methods indicated that the optimal value was K=7. We have updated the fastStructure results to indicate this estimation of K* (lines 351-352). The most obvious difference in genetic structure between K=8 and K=7 is less genetic structure in the UK populations with K=7 (Figure S2).

Reviewers' comments:

Reviewer #1 (Remarks to the Author):

First of all, I would like to thank the authors for taking the time to seriously address my comments. In my opinion, they were well addressed. I find the new version of the manuscript much clearer and more precise.

I do have one remaining major comment though, concerning the comment about the bridgehead hypothesis (comment number 7 in the rebuttal letter). I might have been unclear in my original comment, and I apologize for that. My main concern was that, when I read the manuscript, the major claim I understand is that UK has undergone a multiple introduction demographic history. This point is clearly and convincingly demonstrated.

However, though the bridgehead hypothesis is consistent with historical records, sampling considered in this paper (Germany, Canada and New Zealand) does not seem quite sufficient to confirm a bridgehead dynamic; rather, the results (Figure 2 and 3) only suggest this hypothesis, but, as stated in the discussion (l.508), it only suggests. Therefore, since this hypothesis is relevant in this paper, it should be kept in the Discussion, but it should also be treated accordingly in the rest of the text, namely as a hypothetical token. In other words, it should not be formulated as is in the abstract (l.34), the introduction (l.70 and l.125-126), and especially, in the title. This paper uses population genomic and historical analysis to reveal a multiple introduction history in *Mimulus guttatus*, but only suggest a bridgehead history, a point that warrants further studies. Otherwise, the methods and results sections should be modified to show how this hypothesis is more than just suggested by the data and how it can be strongly and convincingly validated; and only then, earn its place in the title.

Reviewer #2 (Remarks to the Author):

The manuscript addressed by Vallejo-Marin et al. provides a good and solid framework to reconstruct the recent evolutionary history of successive invasion and to understand the evolutionary mechanisms underlying invasiveness.

The revised version is clearer than the previous one and I greatly appreciated the additions in the "Materials and Methods" session, especially about the parameters used for read mapping, SNP calling and filtering.

All of my previous comments have been taken into account by the authors and I find this revised version easier to read and very convincing.

In my opinion, I have no doubt about the great interest to the evolutionary and ecological community and I have no further comments on this revised version.

Response to reviewers

We thank the Editor and both Reviewers for their positive and encouraging assessment of our revised paper. We have now incorporated the suggestion of Reviewer 1, regarding the bridgehead hypothesis.

Reviewer #1

First of all, I would like to thank the authors for taking the time to seriously address my comments. In my opinion, they were well addressed. I find the new version of the manuscript much clearer and more precise.

I do have one remaining major comment though, concerning the comment about the bridgehead hypothesis (comment number 7 in the rebuttal letter). I might have been unclear in my original comment, and I apologize for that. My main concern was that, when I read the manuscript, the major claim I understand is that UK has undergone a multiple introduction demographic history. This point is clearly and convincingly demonstrated.

*However, though the bridgehead hypothesis is consistent with historical records, sampling considered in this paper (Germany, Canada and New Zealand) does not seem quite sufficient to confirm a bridgehead dynamic; rather, the results (Figure 2 and 3) only suggest this hypothesis, but, as stated in the discussion (l.508), it only suggests. Therefore, since this hypothesis is relevant in this paper, it should be kept in the Discussion, but it should also be treated accordingly in the rest of the text, namely as a hypothetical token. In other words, it should not be formulated as is in the abstract (l.34), the introduction (l.70 and l.125-126), and especially, in the title. This paper uses population genomic and historical analysis to reveal a multiple introduction history in *Mimulis guttatus*, but only suggest a bridgehead history, a point that warrants further studies. Otherwise, the methods and results sections should be modified to show how this hypothesis is more than just suggested by the data and how it can be strongly and convincingly validated; and only then, earn its place in the title.*

We thank the Reviewer for this assessment and suggestion. We have embraced the Reviewer's suggestion regarding the presentation of the bridgehead hypothesis, and modified the Title, Abstract, Introduction and Discussion including in all the lines mentioned by the reviewer. We hope the Editor and Reviewer agree that the modifications now fully address this suggestion. The changes we made to the text are highlighted in the manuscript in yellow for ease of reference.

Reviewer #2

No comments to address.